# The Numerics of GANs

**Lars Mescheder**
Autonomous Vision Group
MPI Tübingen
lars.mescheder@tuebingen.mpg.de

**Sebastian Nowozin**
Machine Intelligence and Perception Group
Microsoft Research
sebastian.nowozin@microsoft.com

**Andreas Geiger**
Autonomous Vision Group
MPI Tübingen
andreas.geiger@tuebingen.mpg.de

## Abstract

In this paper, we analyze the numerics of common algorithms for training Generative Adversarial Networks (GANs). Using the formalism of smooth two-player games we analyze the associated gradient vector field of GAN training objectives. Our findings suggest that the convergence of current algorithms suffers due to two factors: i) presence of eigenvalues of the Jacobian of the gradient vector field with zero real-part, and ii) eigenvalues with big imaginary part. Using these findings, we design a new algorithm that overcomes some of these limitations and has better convergence properties. Experimentally, we demonstrate its superiority on training common GAN architectures and show convergence on GAN architectures that are known to be notoriously hard to train.

## 1 Introduction

Generative Adversarial Networks (GANs) [10] have been very successful in learning probability distributions. Since their first appearance, GANs have been successfully applied to a variety of tasks, including image-to-image translation [12], image super-resolution [13], image in-painting [27] domain adaptation [26], probabilistic inference [14, 9, 8] and many more.

While very powerful, GANs are known to be notoriously hard to train. The standard strategy for stabilizing training is to carefully design the model, either by adapting the architecture [21] or by selecting an easy-to-optimize objective function [23, 4, 11].

In this work, we examine the general problem of finding local Nash-equilibria of smooth games. We revisit the de-facto standard algorithm for finding such equilibrium points, simultaneous gradient ascent. We theoretically show that the main factors preventing the algorithm from converging are the presence of eigenvalues of the Jacobian of the associated gradient vector field with zero real-part and eigenvalues with a large imaginary part. The presence of the latter is also one of the reasons that make saddle-point problems more difficult than local optimization problems. Utilizing these insights, we design a new algorithm that overcomes some of these problems. Experimentally, we show that our algorithm leads to stable training on many GAN architectures, including some that are known to be hard to train.

Our technique is orthogonal to strategies that try to make the GAN-game well-defined, e.g. by adding instance noise [24] or by using the Wasserstein-divergence [4, 11]: while these strategies try to ensure the existence of Nash-equilibria, our paper deals with their computation and the numerical difficulties that can arise in practice.

In summary, our contributions are as follows:

- We identify the main reasons why simultaneous gradient ascent often fails to find local Nash-equilibria.
- By utilizing these insights, we design a new, more robust algorithm for finding Nash-equilibria of smooth two-player games.
- We empirically demonstrate that our method enables stable training of GANs on a variety of architectures and divergence measures.

The proofs for the theorems in this paper can be found the supplementary material.[1]

## 2 Background

In this section we first revisit the concept of Generative Adversarial Networks (GANs) from a divergence minimization point of view. We then introduce the concept of a smooth (non-convex) two-player game and define the terminology used in the rest of the paper. Finally, we describe simultaneous gradient ascent, the de-facto standard algorithm for finding Nash-equilibria of such games, and derive some of its properties.

### 2.1 Divergence Measures and GANs

Generative Adversarial Networks are best understood in the context of divergence minimization: assume we are given a divergence function $D$, i.e. a function that takes a pair of probability distributions as input, outputs an element from $[0, \infty]$ and satisfies $D(p, p) = 0$ for all probability distributions $p$. Moreover, assume we are given some target distribution $p_0$ from which we can draw i.i.d. samples and a parametric family of distributions $q_\theta$ that also allows us to draw i.i.d. samples. In practice $q_\theta$ is usually implemented as a neural network that acts on a hidden code $z$ sampled from some known distribution and outputs an element from the target space. Our goal is to find $\bar{\theta}$ that minimizes the divergence $D(p_0, q_\theta)$, i.e. we want to solve the optimization problem

$$\min_\theta D(p_0, q_\theta). \tag{1}$$

Most divergences that are used in practice can be represented in the following form [10, 16, 4]:

$$D(p, q) = \max_{f \in \mathcal{F}} \mathrm{E}_{x \sim q} \left[ g_1(f(x)) \right] - \mathrm{E}_{x \sim p} \left[ g_2(f(x)) \right] \tag{2}$$

for some function class $\mathcal{F} \subseteq \mathcal{X} \to \mathbb{R}$ and convex functions $g_1, g_2 : \mathbb{R} \to \mathbb{R}$. Together with (1), this leads to mini-max problems of the form

$$\min_\theta \max_{f \in \mathcal{F}} \mathrm{E}_{x \sim q_\theta} \left[ g_1(f(x)) \right] - \mathrm{E}_{x \sim p_0} \left[ g_2(f(x)) \right]. \tag{3}$$

These divergences include the Jensen-Shannon divergence [10], all f-divergences [16], the Wasserstein divergence [4] and even the indicator divergence, which is 0 if $p = q$ and $\infty$ otherwise.

In practice, the function class $\mathcal{F}$ in (3) is approximated with a parametric family of functions, e.g. parameterized by a neural network. Of course, when minimizing the divergence w.r.t. this approximated family, we no longer minimize the correct divergence. However, it can be verified that taking any class of functions in (3) leads to a divergence function for appropriate choices of $g_1$ and $g_2$. Therefore, some authors call these divergence functions *neural network divergences* [5].

### 2.2 Smooth Two-Player Games

A differentiable two-player game is defined by two utility functions $f(\phi, \theta)$ and $g(\phi, \theta)$ defined over a common space $(\phi, \theta) \in \Omega_1 \times \Omega_2$. $\Omega_1$ corresponds to the possible actions of player 1, $\Omega_2$ corresponds to the possible actions of player 2. The goal of player 1 is to maximize $f$, whereas player 2 tries to maximize $g$. In the context of GANs, $\Omega_1$ is the set of possible parameter values for the generator, whereas $\Omega_2$ is the set of possible parameter values for the discriminator. We call a game a zero-sum game if $f = -g$. Note that the derivation of the GAN-game in Section 2.1 leads to a zero-sum game,

**Algorithm 1** Simultaneous Gradient Ascent (SimGA)

---

1: **while** not converged **do**
2:     $v_\phi \leftarrow \nabla_\phi f(\theta, \phi)$
3:     $v_\theta \leftarrow \nabla_\theta g(\theta, \phi)$
4:     $\phi \leftarrow \phi + h v_\phi$
5:     $\theta \leftarrow \theta + h v_\theta$
6: **end while**

---

whereas in practice people usually employ a variant of this formulation that is not a zero-sum game for better convergence [10].

Our goal is to find a Nash-equilibrium of the game, i.e. a point $\bar{x} = (\bar{\phi}, \bar{\theta})$ given by the two conditions

$$\bar{\phi} \in \operatorname*{argmax}_\phi f(\phi, \bar{\theta}) \quad \text{and} \quad \bar{\theta} \in \operatorname*{argmax}_\theta g(\bar{\phi}, \theta). \tag{4}$$

We call a point $(\bar{\phi}, \bar{\theta})$ a local Nash-equilibrium, if (4) holds in a local neighborhood of $(\bar{\phi}, \bar{\theta})$.

Every differentiable two-player game defines a vector field

$$v(\phi, \theta) = \begin{pmatrix} \nabla_\phi f(\phi, \theta) \\ \nabla_\theta g(\phi, \theta) \end{pmatrix}. \tag{5}$$

We call $v$ the *associated gradient vector field* to the game defined by $f$ and $g$.

For the special case of zero-sum two-player games, we have $g = -f$ and thus

$$v'(\phi, \theta) = \begin{pmatrix} \nabla_\phi^2 f(\phi, \theta) & \nabla_{\phi,\theta} f(\phi, \theta) \\ -\nabla_{\phi,\theta} f(\phi, \theta) & -\nabla_\theta^2 f(\phi, \theta) \end{pmatrix}. \tag{6}$$

As a direct consequence, we have the following:

**Lemma 1.** *For zero-sum games, $v'(x)$ is negative (semi-)definite if and only if $\nabla_\phi^2 f(\phi, \theta)$ is negative (semi-)definite and $\nabla_\theta^2 f(\phi, \theta)$ is positive (semi-)definite.*

**Corollary 2.** *For zero-sum games, $v'(\bar{x})$ is negative semi-definite for any local Nash-equilibrium $\bar{x}$. Conversely, if $\bar{x}$ is a stationary point of $v(x)$ and $v'(\bar{x})$ is negative definite, then $\bar{x}$ is a local Nash-equilibrium.*

Note that Corollary 2 is not true for general two-player games.

## 2.3 Simultaneous Gradient Ascent

The de-facto standard algorithm for finding Nash-equilibria of general smooth two-player games is Simultaneous Gradient Ascent (SimGA), which was described in several works, for example in [22] and, more recently also in the context of GANs, in [16]. The idea is simple and is illustrated in Algorithm 1. We iteratively update the parameters of the two players by simultaneously applying gradient ascent to the utility functions of the two players. This can also be understood as applying the Euler-method to the ordinary differential equation

$$\frac{\mathrm{d}}{\mathrm{d}t} x(t) = v(x(t)), \tag{7}$$

where $v(x)$ is the associated gradient vector field of the two-player game.

It can be shown that simultaneous gradient ascent converges locally to a Nash-equilibrium for a zero-sum game, if the Hessian of both players is negative definite [16, 22] and the learning rate is small enough. Unfortunately, in the context of GANs the former condition is rarely met. We revisit the properties of simultaneous gradient ascent in Section 3 and also show a more subtle property, namely that even if the conditions for the convergence of simultaneous gradient ascent are met, it might require extremely small step sizes for convergence if the Jacobian of the associated gradient vector field has eigenvalues with large imaginary part.

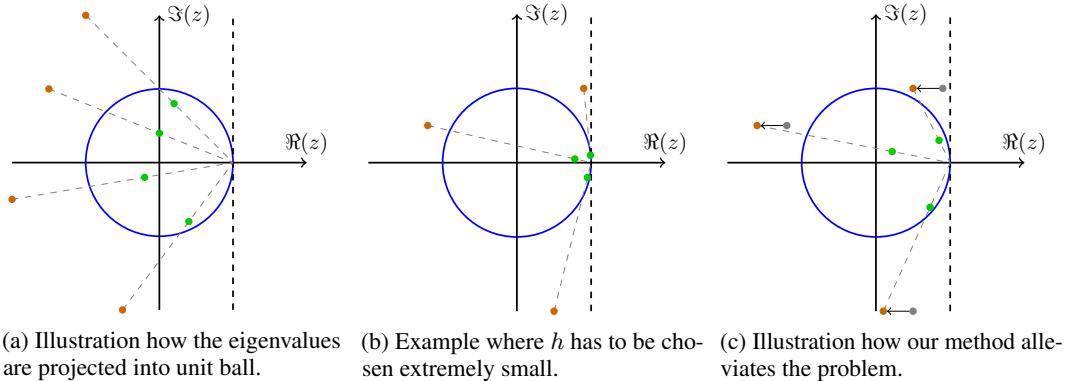

(a) Illustration how the eigenvalues are projected into unit ball.

(b) Example where $h$ has to be chosen extremely small.

(c) Illustration how our method alleviates the problem.

Figure 1: Images showing how the eigenvalues of $A$ are projected into the unit circle and what causes problems: when discretizing the gradient flow with step size $h$, the eigenvalues of the Jacobian at a fixed point are projected into the unit ball along rays from 1. However, this is only possible if the eigenvalues lie in the left half plane and requires extremely small step sizes $h$ if the eigenvalues are close to the imaginary axis. The proposed method moves the eigenvalues to the left in order to make the problem better posed, thus allowing the algorithm to converge for reasonable step sizes.

## 3   Convergence Theory

In this section, we analyze the convergence properties of the most common method for training GANs, simultaneous gradient ascent[2]. We show that two major failure causes for this algorithm are eigenvalues of the Jacobian of the associated gradient vector field with zero real-part as well as eigenvalues with large imaginary part.

For our theoretical analysis, we start with the following classical theorem about the convergence of fixed-point iterations:

**Proposition 3.** *Let $F : \Omega \to \Omega$ be a continuously differential function on an open subset $\Omega$ of $\mathbb{R}^n$ and let $\bar{x} \in \Omega$ be so that*

1. *$F(\bar{x}) = \bar{x}$, and*

2. *the absolute values of the eigenvalues of the Jacobian $F'(\bar{x})$ are all smaller than 1.*

*Then there is an open neighborhood $U$ of $\bar{x}$ so that for all $x_0 \in U$, the iterates $F^{(k)}(x_0)$ converge to $\bar{x}$. The rate of convergence is at least linear. More precisely, the error $\|F^{(k)}(x_0) - \bar{x}\|$ is in $\mathcal{O}(|\lambda_{max}|^k)$ for $k \to \infty$ where $\lambda_{max}$ is the eigenvalue of $F'(\bar{x})$ with the largest absolute value.*

*Proof.*  See [6], Proposition 4.4.1. □

In numerics, we often consider functions of the form

$$F(x) = x + h\, G(x) \tag{8}$$

for some $h > 0$. Finding fixed points of $F$ is then equivalent to finding solutions to the nonlinear equation $G(x) = 0$ for $x$. For $F$ as in (8), the Jacobian is given by

$$F'(x) = I + h\, G'(x). \tag{9}$$

Note that in general neither $F'(x)$ nor $G'(x)$ are symmetric and can therefore have complex eigenvalues.

The following Lemma gives an easy condition, when a fixed point of $F$ as in (8) satisfies the conditions of Proposition 3.

**Lemma 4.** *Assume that $A \in \mathbb{R}^{n \times n}$ only has eigenvalues with negative real-part and let $h > 0$. Then the eigenvalues of the matrix $I + h\,A$ lie in the unit ball if and only if*

$$h < \frac{1}{|\Re(\lambda)|} \frac{2}{1 + \left(\frac{\Im(\lambda)}{\Re(\lambda)}\right)^2} \tag{10}$$

*for all eigenvalues $\lambda$ of $A$.*

**Corollary 5.** *If $v'(\bar{x})$ only has eigenvalues with negative real-part at a stationary point $\bar{x}$, then Algorithm 1 is locally convergent to $\bar{x}$ for $h > 0$ small enough.*

Equation 10 shows that there are two major factors that determine the maximum possible step size $h$: (i) the maximum value of $\Re(\lambda)$ and (ii) the maximum value $q$ of $|\Im(\lambda)/\Re(\lambda)|$. Note that as $q$ goes to infinity, we have to choose $h$ according to $\mathcal{O}(q^{-2})$ which can quickly become extremely small. This is visualized in Figure 1: if $G'(\bar{x})$ has an eigenvalue with small absolute real part but big imaginary part, $h$ needs to be chosen extremely small to still achieve convergence. Moreover, even if we make $h$ small enough, most eigenvalues of $F'(\bar{x})$ will be very close to 1, which leads by Proposition 3 to very slow convergence of the algorithm. This is in particular a problem of simultaneous gradient ascent for two-player games (in contrast to gradient ascent for local optimization), where the Jacobian $G'(\bar{x})$ is not symmetric and can therefore have non-real eigenvalues.

## 4    Consensus Optimization

In this section, we derive the proposed method and analyze its convergence properties.

### 4.1    Derivation

Finding stationary points of the vector field $v(x)$ is equivalent to solving the equation $v(x) = 0$. In the context of two-player games this means solving the two equations

$$\nabla_\phi f(\phi, \theta) = 0 \quad \text{and} \quad \nabla_\theta g(\phi, \theta) = 0. \tag{11}$$

A simple strategy for finding such stationary points is to minimize $L(x) = \frac{1}{2}\|v(x)\|^2$ for $x$. Unfortunately, this can result in unstable stationary points of $v$ or other local minima of $\frac{1}{2}\|v(x)\|^2$ and in practice, we found it did not work well.

We therefore consider a modified vector field $w(x)$ that is as close as possible to the original vector field $v(x)$, but at the same time still minimizes $L(x)$ (at least locally). A sensible candidate for such a vector field is

$$w(x) = v(x) - \gamma \nabla L(x) \tag{12}$$

for some $\gamma > 0$. A simple calculation shows that the gradient $\nabla L(x)$ is given by

$$\nabla L(x) = v'(x)^\mathsf{T} v(x). \tag{13}$$

This vector field is the gradient vector field associated to the modified two-player game given by the two modified utility functions

$$\tilde{f}(\phi, \theta) = f(\phi, \theta) - \gamma L(\phi, \theta) \quad \text{and} \quad \tilde{g}(\phi, \theta) = g(\phi, \theta) - \gamma L(\phi, \theta). \tag{14}$$

The regularizer $L(\phi, \theta)$ encourages agreement between the two players. Therefore we call the resulting algorithm *Consensus Optimization* (Algorithm 2). [3] [4]

**Algorithm 2** Consensus optimization

1: **while** not converged **do**
2:     $v_\phi \leftarrow \nabla_\phi(f(\theta, \phi) - \gamma L(\theta, \phi))$
3:     $v_\theta \leftarrow \nabla_\theta(g(\theta, \phi) - \gamma L(\theta, \phi))$
4:     $\phi \leftarrow \phi + h v_\phi$
5:     $\theta \leftarrow \theta + h v_\theta$
6: **end while**

## 4.2 Convergence

For analyzing convergence, we consider a more general algorithm than in Section 4.1 which is given by iteratively applying a function $F$ of the form

$$F(x) = x + h\,A(x)v(x). \tag{15}$$

for some step size $h > 0$ and an invertible matrix $A(x)$ to $x$. Consensus optimization is a special case of this algorithm for $A(x) = I - \gamma\,v'(x)^\mathsf{T}$. We assume that $\frac{1}{\gamma}$ is not an eigenvalue of $v'(x)^\mathsf{T}$ for any $x$, so that $A(x)$ is indeed invertible.

**Lemma 6.** *Assume $h > 0$ and $A(x)$ invertible for all $x$. Then $\bar{x}$ is a fixed point of* (15) *if and only if it is a stationary point of $v$. Moreover, if $\bar{x}$ is a stationary point of $v$, we have*

$$F'(\bar{x}) = I + hA(\bar{x})v'(\bar{x}). \tag{16}$$

**Lemma 7.** *Let $A(x) = I - \gamma v'(x)^\mathsf{T}$ and assume that $v'(\bar{x})$ is negative semi-definite and invertible[5]. Then $A(\bar{x})v'(\bar{x})$ is negative definite.*

As a consequence of Lemma 6 and Lemma 7, we can show local convergence of our algorithm to a local Nash equilibrium:

**Corollary 8.** *Let $v(x)$ be the associated gradient vector field of a two-player zero-sum game and $A(x) = I - \gamma v'(x)^\mathsf{T}$. If $\bar{x}$ is a local Nash-equilibrium, then there is an open neighborhood $U$ of $\bar{x}$ so that for all $x_0 \in U$, the iterates $F^{(k)}(x_0)$ converge to $\bar{x}$ for $h > 0$ small enough.*

Our method solves the problem of eigenvalues of the Jacobian with (approximately) zero real-part. As the next Lemma shows, it also alleviates the problem of eigenvalues with a big imaginary-to-real-part-quotient:

**Lemma 9.** *Assume that $A \in \mathbb{R}^{n \times n}$ is negative semi-definite. Let $q(\gamma)$ be the maximum of $\frac{|\Im(\lambda)|}{|\Re(\lambda)|}$ (possibly infinite) with respect to $\lambda$ where $\lambda$ denotes the eigenvalues of $A - \gamma A^\mathsf{T} A$ and $\Re(\lambda)$ and $\Im(\lambda)$ denote their real and imaginary part respectively. Moreover, assume that $A$ is invertible with $|Av| \geq \rho|v|$ for $\rho > 0$ and let*

$$c = \min_{v \in \mathbb{S}(\mathbb{C}^n)} \frac{|\bar{v}^\mathsf{T}(A + A^\mathsf{T})v|}{|\bar{v}^\mathsf{T}(A - A^\mathsf{T})v|} \tag{17}$$

*where $\mathbb{S}(\mathbb{C}^n)$ denotes the unit sphere in $\mathbb{C}^n$. Then*

$$q(\gamma) \leq \frac{1}{c + 2\rho^2\gamma}. \tag{18}$$

Lemma 9 shows that the imaginary-to-real-part-quotient can be made arbitrarily small for an appropriate choice of $\gamma$. According to Proposition 3, this leads to better convergence properties near a local Nash-equilibrium.

## 5 Experiments

**Mixture of Gaussians**   In our first experiment we evaluate our method on a simple $2D$-example where our goal is to learn a mixture of 8 Gaussians with standard deviations equal to $10^{-2}$ and modes

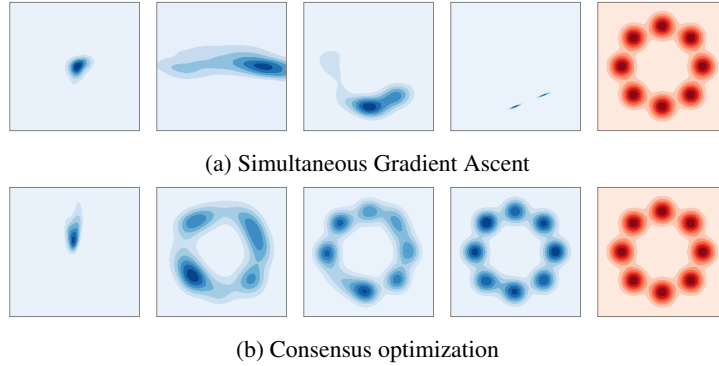

(a) Simultaneous Gradient Ascent

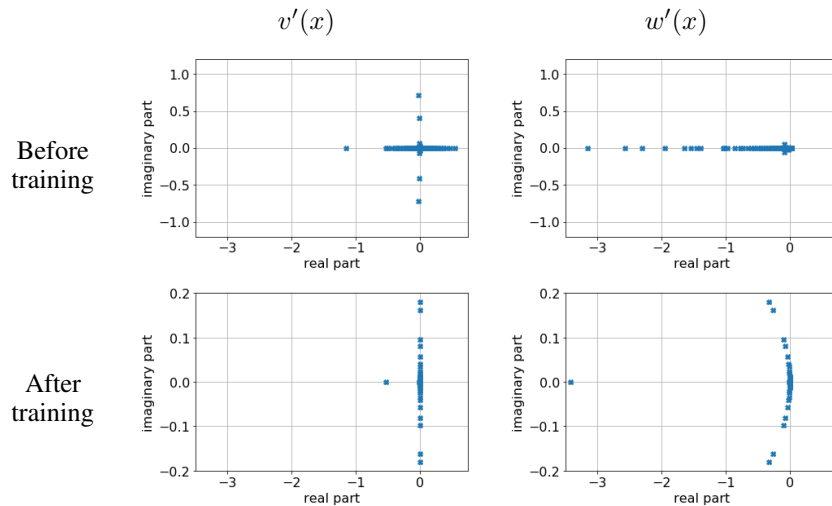

(b) Consensus optimization

Figure 2: Comparison of Simultaneous Gradient Ascent and Consensus optimization on a circular mixture of Gaussians. The images depict from left to right the resulting densities of the algorithm after 0, 5000, 10000 and 20000 iterations as well as the target density (in red).

$$v'(x) \qquad w'(x)$$

Figure 3: Empirical distribution of eigenvalues before and after training using consensus optimization. The first column shows the distribution of the eigenvalues of the Jacobian $v'(x)$ of the unmodified vector field $v(x)$. The second column shows the eigenvalues of the Jacobian $w'(x)$ of the regularized vector field $w(x) = v(x) - \gamma \nabla L(x)$ used in consensus optimization. We see that $v'(x)$ has eigenvalues close to the imaginary axis near the Nash-equilibrium. As predicted theoretically, this is not the case for the regularized vector field $w(x)$. For visualization purposes, the real part of the spectrum of $w'(x)$ before training was clipped.

uniformly distributed around the unit circle. While simplistic, algorithms training GANs often fail to converge even on such simple examples without extensive fine-tuning of the architecture and hyper parameters [15].

For both the generator and critic we use fully connected neural networks with 4 hidden layers and 16 hidden units in each layer. For all layers, we use RELU-nonlinearities. We use a 16-dimensional Gaussian prior for the latent code $z$ and set up the game between the generator and critic using the utility functions as in [10]. To test our method, we run both SimGA and our method with RMSProp and a learning rate of $10^{-4}$ for 20000 steps. For our method, we use a regularization parameter of $\gamma = 10$.

The results produced by SimGA and our method for 0, 5000, 10000 and 20000 iterations are depicted in Figure 2. We see that while SimGA jumps around the modes of the distribution and fails to converge , our method converges smoothly to the target distribution (shown in red). Figure 3 shows the empirical distribution of the eigenvalues of the Jacobian of $v(x)$ and the regularized vector field $w(x)$. It can be seen that near the Nash-equilibrium most eigenvalues are indeed very close to the

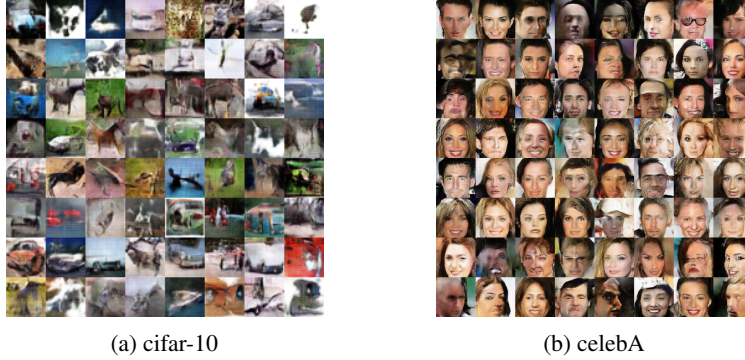

| (a) cifar-10 | (b) celebA |

Figure 4: Samples generated from a model where both the generator and discriminator are given as in [21], but without batch-normalization. For celebA, we also use a constant number of filters in each layer and add additional RESNET-layers.

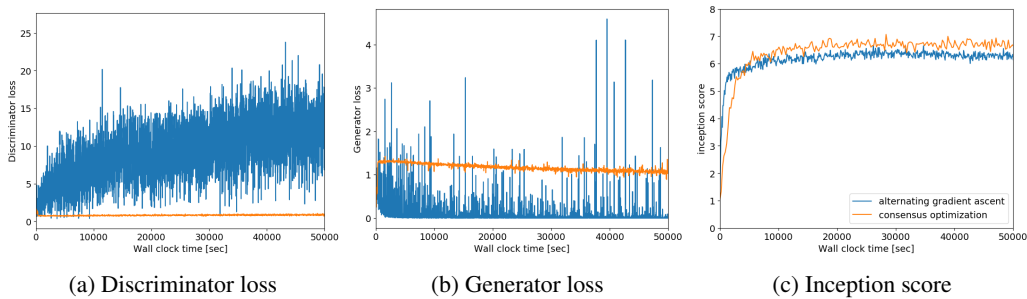

| (a) Discriminator loss | (b) Generator loss | (c) Inception score |

Figure 5: (a) and (b): Comparison of the generator and discriminator loss on a DC-GAN architecture with 3 convolutional layers trained on cifar-10 for consensus optimization (without batch-normalization) and alternating gradient ascent (with batch-normalization). We observe that while alternating gradient ascent leads to highly fluctuating losses, consensus optimization successfully stabilizes the training and makes the losses almost constant during training. (c): Comparison of the inception score over time which was computed using 6400 samples. We see that on this architecture both methods have comparable rates of convergence and consensus optimization achieves slightly better end results.

imaginary axis and that the proposed modification of the vector field used in consensus optimization moves the eigenvalues to the left.

**CIFAR-10 and CelebA** In our second experiment, we apply our method to the cifar-10 and celebA-datasets, using a DC-GAN-like architecture [21] without batch normalization in the generator or the discriminator. For celebA, we additionally use a constant number of filters in each layer and add additional RESNET-layers. These architectures are known to be hard to optimize using simultaneous (or alternating) gradient ascent [21, 4].

Figure 4a and 4b depict samples from the model trained with our method. We see that our method successfully trains the models and we also observe that unlike when using alternating gradient ascent, the generator and discriminator losses remain almost constant during training. This is illustrated in Figure 5. For a quantitative evaluation, we also measured the inception-score [23] over time (Figure 5c), showing that our method compares favorably to a DC-GAN trained with alternating gradient ascent. The improvement of consensus optimization over alternating gradient ascent is even more significant if we use 4 instead of 3 convolutional layers, see Figure 11 in the supplementary material for details.

Additional experimental results can be found in the supplementary material.

## 6 Discussion

While we could prove local convergence of our method in Section 4, we believe that even more insights can be gained by examining global convergence properties. In particular, our analysis from

Section 4 cannot explain why the generator and discriminator losses remain almost constant during training.

Our theoretical results assume the existence of a Nash-equilibrium. When we are trying to minimize an f-divergence and the dimensionality of the generator distribution is misspecified, this might not be the case [3]. Nonetheless, we found that our method works well in practice and we leave a closer theoretical investigation of this fact to future research.

In practice, our method can potentially make formerly instable stationary points of the gradient vector field stable if the regularization parameter is chosen to be high. This may lead to poor solutions. We also found that our method becomes less stable for deeper architectures, which we attribute to the fact that the gradients can have very different scales in such architectures, so that the simple L2-penalty from Section 4 needs to be rescaled accordingly.

Our method can be regarded as an approximation to the implicit Euler method for integrating the gradient vector field. It can be shown that the implicit Euler method has appealing stability properties [7] that can be translated into convergence theorems for local Nash-equilibria. However, the implicit Euler method requires the solution of a nonlinear equation in each iteration. Nonetheless, we believe that further progress can be made by finding better approximations to the implicit Euler method.

An alternative interpretation is to view our method as a second order method. We hence believe that further progress can be made by revisiting second order optimization methods [2, 18] in the context of saddle point problems.

# 7  Related Work

Saddle point problems do not only arise in the context of training GANs. For example, the popular actor-critic models [20] in reinforcement learning are also special cases of saddle-point problems.

Finding a stable algorithm for training GANs is a long standing problem and multiple solutions have been proposed. Unrolled GANs [15] unroll the optimization with respect to the critic, thereby giving the generator more informative gradients. Though unrolling the optimization was shown to stabilize training, it can be cumbersome to implement and in addition it also results in a big model. As was recently shown, the stability of GAN-training can be improved by using objectives derived from the Wasserstein-1-distance (induced by the Kantorovich-Rubinstein-norm) instead of f-divergences [4, 11]. While Wasserstein-GANs often provide a good solution for the stable training of GANs, they require keeping the critic optimal, which can be time-consuming and can in practice only be achieved approximately, thus violating the conditions for theoretical guarantees. Moreover, some methods like Adversarial Variational Bayes [14] explicitly prescribe the divergence measure to be used, thus making it impossible to apply Wasserstein-GANs. Other approaches that try to stabilize training, try to design an easy-to-optimize architecture [23, 21] or make use of additional labels [23, 17].

In contrast to all the approaches described above, our work focuses on stabilizing training on a wide range of architecture and divergence functions.

# 8  Conclusion

In this work, starting from GAN objective functions we analyzed the general difficulties of finding local Nash-equilibria in smooth two-player games. We pinpointed the major numerical difficulties that arise in the current state-of-the-art algorithms and, using our insights, we presented a new algorithm for training generative adversarial networks. Our novel algorithm has favorable properties in theory and practice: from the theoretical viewpoint, we showed that it is locally convergent to a Nash-equilibrium even if the eigenvalues of the Jacobian are problematic. This is particularly interesting for games that arise in the context of GANs where such problems are common. From the practical viewpoint, our algorithm can be used in combination with any GAN-architecture whose objective can be formulated as a two-player game to stabilize the training. We demonstrated experimentally that our algorithm stabilizes the training and successfully combats training issues like mode collapse. We believe our work is a first step towards an understanding of the numerics of GAN training and more general deep learning objective functions.

## Acknowledgements

This work was supported by Microsoft Research through its PhD Scholarship Programme.

## Footnotes

[1]The code for all experiments in this paper is available under `https://github.com/LMescheder/TheNumericsOfGANs`.

[2]A similar analysis of alternating gradient ascent, a popular alternative to simultaneous gradient ascent, can be found in the supplementary material.

[3]This algorithm requires backpropagation through the squared norm of the gradient with respect to the weights of the network. This is sometimes called *double backpropagation* and is for example supported by the deep learning frameworks Tensorflow [1] and PyTorch [19].

[4]As was pointed out by Ferenc Huszár in one of his blog posts on `www.inference.vc`, naively implementing this algorithm in a mini-batch setting leads to biased estimates of $L(x)$. However, the bias goes down linearly with the batch size, which justifies the usage of consensus optimization in a mini-batch setting. Alternatively, it is possible to debias the estimate by subtracting a multiple of the sample variance of the gradients, see the supplementary material for details.

[5]Note that $v'(\bar{x})$ is usually not symmetric and therefore it is possible that $v'(\bar{x})$ is negative semi-definite and invertible but not negative-definite.

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
