[Supplementary Material]

# The Numerics of GANs: Supplementary Material

**Lars Mescheder**
Autonomous Vision Group
MPI Tübingen
lars.mescheder@tuebingen.mpg.de

**Sebastian Nowozin**
Machine Intelligence and Perception Group
Microsoft Research
sebastian.nowozin@microsoft.com

**Andreas Geiger**
Autonomous Vision Group
MPI Tübingen
andreas.geiger@tuebingen.mpg.de

## Abstract

This document contains proofs that were omitted in the main text of the paper "The Numerics of GANs" as well as additional theoretical results. We also include additional experimental results and demonstrate that our method leads to stable training of GANs on a variety of architectures and divergence measures.

## Proofs

This section contains proofs that were omitted in the main text.

### Smooth two player games

**Lemma 1.** *For zero-sum games, $v'(x)$ is negative (semi-)definite if and only if $\nabla_\phi^2 f(\phi, \theta)$ is negative (semi-)definite and $\nabla_\theta^2 f(\phi, \theta)$ is positive (semi-)definite.*

*Proof.* We have for any $w = (w_1, w_2) \neq 0$

$$w^\mathsf{T} v'(x) w = w_1^\mathsf{T} \nabla_\phi^2 f(\phi, \theta) w_1 - w_2^\mathsf{T} \nabla_\theta^2 f(\phi, \theta) w_2. \tag{19}$$

Hence, we have $w^\mathsf{T} v'(x) w < 0$ for all vectors $w \neq 0$ if and only if $w_1^\mathsf{T} \nabla_\phi^2 f(\phi, \theta) w_1 < 0$ and $w_2^\mathsf{T} \nabla_\theta^2 f(\phi, \theta) w_2 > 0$ for all vectors $w_1, w_2 \neq 0$.

This shows that $v'(x)$ is negative definite if and only if $\nabla_\phi^2 f(\phi, \theta)$ is negative definite and $\nabla_\theta^2 f(\phi, \theta)$ is positive definite.

A similar proof shows that $v'(x)$ is negative semi-definite if and only if $\nabla_\phi^2 f(\phi, \theta)$ is negative semi-definite and $\nabla_\theta^2 f(\phi, \theta)$ is positive semi-definite. $\qquad \square$

**Corollary 2.** *For zero-sum games, $v'(\bar{x})$ is negative semi-definite for any local Nash-equilibrium $\bar{x}$. Conversely, if $\bar{x}$ is a stationary point of $v(x)$ and $v'(\bar{x})$ is negative definite, then $\bar{x}$ is a local Nash-equilibrium.*

*Proof.* If $\bar{x}$ is a local Nash-equilibrium, $\nabla_\phi^2 f(\bar{\phi}, \bar{\theta})$ is negative semi-definite and $\nabla_\theta^2 f(\bar{\phi}, \bar{\theta})$ is positive semi-definite, so $v'(\bar{x})$ is negative definite by Lemma 1.

Conversely, if $v'(\bar{x})$ is negative definite, $\nabla_\phi^2 f(\bar{\phi}, \bar{\theta})$ is negative definite and $\nabla_\theta^2 f(\bar{\phi}, \bar{\theta})$ positive definite by Lemma 1. This implies that $\bar{x}$ is a local Nash-equilibrium of the two-player game defined by $f$. $\qquad \square$

## Convergence theory

**Proposition 3.** *Let $F : \Omega \to \Omega$ be a continuously differential function on an open subset $\Omega$ of $\mathbb{R}^n$ and let $\bar{x} \in \Omega$ be so that*

1. *$F(\bar{x}) = \bar{x}$, and*

2. *the absolute values of the eigenvalues of the Jacobian $F'(\bar{x})$ are all smaller than 1.*

*Then there is an open neighborhood $U$ of $\bar{x}$ so that for all $x_0 \in U$, the iterates $F^{(k)}(x_0)$ converge to $\bar{x}$. The rate of convergence is at least linear. More precisely, the error $\|F^{(k)}(x_0) - \bar{x}\|$ is in $\mathcal{O}(|\lambda_{max}|^k)$ for $k \to \infty$ where $\lambda_{max}$ is the eigenvalue of $F'(\bar{x})$ with the largest absolute value.*

*Proof.* See [6], Proposition 4.4.1. $\qquad\square$

**Lemma 4.** *Assume that $A \in \mathbb{R}^{n \times n}$ only has eigenvalues with negative real-part and let $h > 0$. Then the eigenvalues of the matrix $I + h\,A$ lie in the unit ball if and only if*

$$h < \frac{1}{|\Re(\lambda)|} \frac{2}{1 + \left(\frac{\Im(\lambda)}{\Re(\lambda)}\right)^2} \tag{10}$$

*for all eigenvalues $\lambda$ of $A$.*

*Proof.* For $\lambda = -a + b\,i$ with $a > 0$ we have

$$|1 + h\,\lambda|^2 = (1 - h\,a)^2 + h^2\,b^2 = 1 + h^2\,b^2 + h^2\,a^2 - 2\,h\,a. \tag{20}$$

For $h > 0$, this is smaller than 1 if and only if

$$h < \frac{2\,a}{b^2 + a^2} = \frac{2a^{-1}}{(b/a)^2 + 1}. \tag{21}$$

$\qquad\square$

**Corollary 5.** *If $v'(\bar{x})$ only has eigenvalues with negative real-part at a stationary point $\bar{x}$, then Algorithm 1 is locally convergent to $\bar{x}$ for $h > 0$ small enough.*

*Proof.* This is a direct consequence of Proposition 3 and Lemma 4. $\qquad\square$

**Lemma 6.** *Assume $h > 0$ and $A(x)$ invertible for all $x$. Then $\bar{x}$ is a fixed point of (15) if and only if it is a stationary point of $v$. Moreover, if $\bar{x}$ is a stationary point of $v$, we have*

$$F'(\bar{x}) = I + hA(\bar{x})v'(\bar{x}). \tag{16}$$

*Proof.* If $v(\bar{x}) = 0$, then $F(\bar{x}) = \bar{x}$, so $\bar{x}$ is a fixed point of $F$. Conversely, if $\bar{x}$ satisfies $F(\bar{x}) = \bar{x}$, we have $A(\bar{x})\,v(\bar{x}) = 0$. Because we assume $A(\bar{x})$ to be invertible, this shows $v(\bar{x}) = 0$.

Now, the $i^{th}$ partial derivative of $F(x)$ is given by

$$\partial_{x_i} F(x) = b_i + h\,\partial_{x_i} A(x)\,v(x) + hA(x)\partial_{x_i} v(x) \tag{22}$$

where $b_i$ denotes the $i^{th}$ unit basis vector. For a fixed point $\bar{x}$ we have by the first part of the proof $v(\bar{x}) = 0$ and therefore

$$\partial_{x_i} F(\bar{x}) = b_i + hA(\bar{x})\partial_{x_i} v(\bar{x}). \tag{23}$$

This shows

$$F'(\bar{x}) = I + hA(\bar{x})\,v'(\bar{x}). \tag{24}$$

$\qquad\square$

**Lemma 7.** *Let $A(x) = I - \gamma v'(x)^{\mathsf{T}}$ and assume that $v'(\bar{x})$ is negative semi-definite and invertible[6]. Then $A(\bar{x})v'(\bar{x})$ is negative definite.*

*Proof.* We have for all $w \neq 0$:

$$w^\mathsf{T} A(\bar{x}) v'(\bar{x}) w = w^\mathsf{T} v'(\bar{x}) w - \gamma \|v'(\bar{x}) w\|^2 \leq -\gamma \|v'(\bar{x}) w\|^2 < 0 \tag{25}$$

as $v'(\bar{x}) w \neq 0$ for $w \neq 0$. $\qquad \square$

**Corollary 8.** *Let $v(x)$ be the associated gradient vector field of a two-player zero-sum game and $A(x) = I - \gamma v'(x)^\mathsf{T}$. If $\bar{x}$ is a local Nash-equilibrium, then there is an open neighborhood $U$ of $\bar{x}$ so that for all $x_0 \in U$, the iterates $F^{(k)}(x_0)$ converge to $\bar{x}$ for $h > 0$ small enough.*

*Proof.* This is a direct consequence of Proposition 3, Lemma 6 and Lemma 7. $\qquad \square$

**Lemma 9.** *Assume that $A \in \mathbb{R}^{n \times n}$ is negative semi-definite. Let $q(\gamma)$ be the maximum of $\frac{|\Im(\lambda)|}{|\Re(\lambda)|}$ (possibly infinite) with respect to $\lambda$ where $\lambda$ denotes the eigenvalues of $A - \gamma A^\mathsf{T} A$ and $\Re(\lambda)$ and $\Im(\lambda)$ denote their real and imaginary part respectively. Moreover, assume that $A$ is invertible with $|Av| \geq \rho |v|$ for $\rho > 0$ and let*

$$c = \min_{v \in \mathbb{S}(\mathbb{C}^n)} \frac{|\bar{v}^\mathsf{T}(A + A^\mathsf{T})v|}{|\bar{v}^\mathsf{T}(A - A^\mathsf{T})v|} \tag{17}$$

*where $\mathbb{S}(\mathbb{C}^n)$ denotes the unit sphere in $\mathbb{C}^n$. Then*

$$q(\gamma) \leq \frac{1}{c + 2\rho^2 \gamma}. \tag{18}$$

*Proof.* Let $v \in \mathbb{C}^n \setminus \{0\}$ be any eigenvector of $B := A - \gamma A^\mathsf{T} A$ and $\lambda \in \mathbb{C}$ the corresponding eigenvalue. We can assume w.l.o.g. that $\|v\| = 1$.

Then

$$\lambda = \lambda \bar{v}^\mathsf{T} v = \bar{v}^\mathsf{T} B v. \tag{26}$$

This implies that

$$\Re(\lambda) = \frac{\lambda + \bar{\lambda}}{2} = \frac{1}{2} \bar{v}^\mathsf{T} \left(B + B^\mathsf{T}\right) v \tag{27}$$

and, similarly,

$$\Im(\lambda) = \frac{\lambda - \bar{\lambda}}{2i} = \frac{1}{2i} \bar{v}^\mathsf{T} \left(B - B^\mathsf{T}\right) v. \tag{28}$$

Consequently, we have

$$\left| \frac{\Im(\lambda)}{\Re(\lambda)} \right| = \frac{|v^\mathsf{T} \left(B - B^\mathsf{T}\right) v|}{|\bar{v}^\mathsf{T} \left(B + B^\mathsf{T}\right) v|} \tag{29}$$

and thus

$$q(\gamma) \leq \max_{v \in \mathbb{S}(\mathbb{C}^n)} \frac{|v^\mathsf{T} \left(B - B^\mathsf{T}\right) v|}{|\bar{v}^\mathsf{T} \left(B + B^\mathsf{T}\right) v|}. \tag{30}$$

However, we have

$$B - B^\mathsf{T} = A - A^\mathsf{T} \quad \text{and} \quad B + B^\mathsf{T} = A + A^\mathsf{T} - 2\gamma A^\mathsf{T} A. \tag{31}$$

This implies, because $A$ is negative semi-definite, that

$$\frac{|v^\mathsf{T} \left(B - B^\mathsf{T}\right) v|}{|\bar{v}^\mathsf{T} \left(B + B^\mathsf{T}\right) v|} = \frac{|v^\mathsf{T} \left(A - A^\mathsf{T}\right) v|}{|\bar{v}^\mathsf{T} \left(A + A^\mathsf{T}\right) v| + 2\gamma \|Av\|^2}. \tag{32}$$

Because $\|Av\|^2 \geq \rho \|v\|^2 = \rho$ this implies the assertion. $\qquad \square$

## Additional Theoretical Results

This section contains some additional theoretical results. On the one hand, we demonstrate how the convergence of gradient ascent and common modifications like momentum and gradient rescaling can be analyzed naturally using Proposition 3. On the other hand, we analyze alternating gradient ascent for two-player games and show that for small step sizes $h > 0$ it is locally convergent towards a Nash-equilibrium if all the eigenvalues of the Jacobian of the associated gradient vector field have negative real-part. We also discuss the bias of consensus optimization in a mini-batch setting and present a possible way to debias it.

**Gradient Ascent**

Let
$$v(x) = \nabla f(x). \tag{33}$$
Then $v'(x) = \nabla^2 f(x)$ is the Hessian-matrix of $f$ at $x$. Let
$$F(x) = x + h\, v(x). \tag{34}$$
A direct consequence of Proposition 3 is

**Proposition 10.** *Any fixed point of* (34) *is a stationary point of the gradient vector field* $v(x)$. *Moreover, if* $\nabla^2 f(x)$ *is negative definite, the fixed point iteration defined by $F$ is locally convergent towards $\bar{x}$ for small $h > 0$.*

*Proof.* This follows directly from Proposition 3. $\qquad\square$

**Momentum**

Let $v(x)$ be a vector field. Using momentum, the operator $F$ can be written as
$$F(x, m) = (x, m) + h\, G(x, m) \tag{35}$$
with $G(x, m) = (m, v(x) - \gamma m)$. The Jacobian of $G$ is given by
$$G'(x, m) = \begin{pmatrix} 0 & I \\ v'(x) & -\gamma I \end{pmatrix}. \tag{36}$$

**Lemma 11.** $\lambda$ *is an eigenvalue of* $G'(x, m)$ *if and only if* $\lambda(\lambda + \gamma)$ *is an eigenvalue of* $v'(x)$.

*Proof.* Let $(w_1, w_2)$ be an eigenvector of $G'(x, m)$ as in (36) with associated eigenvalue $\lambda$. Then
$$\lambda w_1 = w_2 \tag{37}$$
$$\lambda w_2 = v'(\bar{x})w_1 - \gamma w_2, \tag{38}$$
showing that $\lambda(\lambda+\gamma)w_1 = v'(\bar{x})w_1$. As $w_1 = 0$ implies $w_2 = 0$, this shows that $w_1$ is an eigenvector of $v'(\bar{x})$ with associated eigenvalue $\lambda(\lambda + \gamma)$.

Conversely, let $\lambda(\lambda + \gamma)$ be an eigenvalue of $v'(x)$ to the eigenvector $w$. We have
$$\begin{pmatrix} 0 & I \\ v'(x) & -\gamma I \end{pmatrix} \begin{pmatrix} w \\ \lambda w \end{pmatrix} = \begin{pmatrix} \lambda w \\ \lambda(\lambda + \gamma)w - \gamma \lambda w \end{pmatrix} = \lambda \begin{pmatrix} w \\ \lambda w \end{pmatrix} \tag{39}$$
showing that $\lambda$ is an eigenvalue of $G'(x, m)$ to the eigenvector $(w, \lambda w)$. $\qquad\square$

**Corollary 12.** *Any fixed point* $(\bar{x}, \bar{m})$ *of* (35) *satisfies* $\bar{m} = 0$ *and* $v(\bar{x}) = 0$. *Moreover, assume that* $\gamma > 0$ *and that* $v'(\bar{x})$ *only has real negative eigenvalues. Then the fixed point iteration defined by* (35) *is locally convergent towards* $(\bar{x}, \bar{m})$ *for small $h > 0$.*

*Proof.* It is easy to see that any fixed point of (35) must satisfy $\bar{m} = 0$ and $v(\bar{x}) = 0$. If all eigenvalues $\mu$ of $v'(\bar{x})$ are real and non-positive, then all solutions to $\lambda(\lambda + \gamma) = \mu$ have negative real part, showing local convergence by Proposition 3. $\qquad\square$

Note that the proof of Corollary 12 breaks down if $v'(\bar{x})$ has complex eigenvalues, as it is often the case for the associated gradient vector field of two-player games.

**Gradient Rescaling**

In this section we investigate the effect of gradient rescaling as used in ADAM and RMSProp on local convergence. In particular, let
$$F(x, \beta) = \begin{pmatrix} x + \frac{h}{\sqrt{\beta}+\epsilon} v(x) \\ (1 - \alpha)\beta + \alpha\|v(x)\|^2 \end{pmatrix} \tag{40}$$
for some $\alpha > 0$. The Jacobian of (40) is then given by
$$F'(x, \beta) = \begin{pmatrix} I + \frac{h}{\sqrt{\beta}+\epsilon} v'(x) & -\frac{h}{2(\beta+\epsilon)^{3/2}} v(x) \\ \alpha\, v'(x)^\mathsf{T} v(x) & 1 - \alpha \end{pmatrix}. \tag{41}$$

---

**Algorithm 3** Alternating Gradient Ascent

---

1: **while** not converged **do**
2:     $\phi \leftarrow \phi + h \nabla_\phi f(\theta, \phi)$
3:     $\theta \leftarrow \theta + h \nabla_\theta g(\theta, \phi)$
4: **end while**

---

**Proposition 13.** *Any fixed point $(\bar{x}, \bar{\beta})$ of (40) satisfies $\bar{\beta} = 0$ and $v(\bar{x}) = 0$. Moreover, assume that $\alpha \in (0, 1)$ and that all eigenvalues of $I + \frac{h}{\sqrt{\epsilon}} v'(\bar{x})$ lie in the unit ball. Then the fixed point iteration defined by (40) is locally convergent towards $(\bar{x}, \bar{\beta})$.*

*Proof.* It is easy to see that any fixed point $(\bar{x}, \bar{\beta})$ of (40) must satisfy $\bar{\beta} = 0$ and $v(\bar{x}) = 0$. We therefore have

$$F'(\bar{x}, \bar{\beta}) = \begin{pmatrix} I + \frac{h}{\sqrt{\epsilon}} v'(x) & 0 \\ 0 & 1 - \alpha \end{pmatrix}. \tag{42}$$

The eigenvalues of (42) are just the eigenvalues of $I + \frac{h}{\sqrt{\epsilon}} v'(x)$ and $1 - \alpha$ which all lie in the unit ball by assumption. $\square$

**Alternating Gradient Ascent**

Alternating Gradient Ascent (AltGA) applies gradient ascent-updates for the two players in an alternating fashion, see Algorithm 3. For a theoretical analysis, we more generally regard fixed-point methods that iteratively apply a function of the form

$$F(x) = F_2(F_1(x)) \tag{43}$$

to $x$. By the chain rule, the Jacobian of $F$ at $x$ is given by

$$F'(x) = F_2'(F_1(x))F_1'(x). \tag{44}$$

Assume now that $F_1(x) = x + hG_1(x)$ and $F_2(x) = x + hG_2(x)$. Then, if $\bar{x}$ is a fixed point of both $F_1$ and $F_2$, we have

$$F'(\bar{x}) = (I + hG_2'(\bar{x}))(I + hG_1'(\bar{x})) = I + hG_1'(\bar{x}) + hG_2'(\bar{x}) + h^2 G_2'(\bar{x})G_1'(\bar{x}). \tag{45}$$

Let $A(x) := G_2'(x) + G_1'(x)$. Notice that for alternating gradient ascent, $A(x)$ is equal to the Jacobian $v'(x)$ of the gradient vector field $v(x)$. The eigenvalues of $\tilde{A}_h(x) := A(x) + hG_2'(x)G_1'(x)$ depend continuously on $h$. Hence, for $h$ small enough, all eigenvalues of $\tilde{A}_h(x)$ will be arbitrarily close to the eigenvalues of $A(x)$.

As a consequence, we the following analogue of Corollary 5:

**Proposition 14.** *If the Jacobian $v'(\bar{x})$ of the gradient vector field $v(x)$ at a fixed point $\bar{x}$ only has eigenvalues with negative real part, Algorithm 3 is locally convergent to $\bar{x}$ for $h > 0$ small enough.*

*Proof.* This follows directly from the derivation above and Proposition 3. $\square$

**The bias in consensus optimization**

To calculate $\nabla L(x)$ in consensus optimization, we need estimates of $L(x)$, which is defined as

$$L(x) := \frac{1}{2}\|v(x)\|^2. \tag{46}$$

In a mini-batch setting, we can obtain unbiased estimates of $v(x)$ by averaging all gradients $v_i(x)$ for the examples in the mini-batch:

$$v(x) \approx v_B(x) = \frac{1}{B}\sum_{i=1}^{B} v_i(x). \tag{47}$$

However, when we substitute this estimate for $v(x)$ in the definition of $L(x)$, we obtain a biased estimator of $L(x)$ as the next lemma shows. The lemma also gives an explicit expression for the bias.

**Lemma 15.** *Assume that $v_i(x)$, $i = 1, \ldots, B$, are independent and identically distributed unbiased estimates of $v(x)$, i.e. $\mathrm{E}[v_i(x)] = v(x)$ for all $i$. Let*

$$L_B(x) := \frac{1}{2}\Big\|\frac{1}{B}\sum_{i=1}^{B} v_i(x)\Big\|^2. \tag{48}$$

*Then $L_B(x)$ is a biased estimator of $L(x)$ with*

$$\mathrm{E}[L_B(x)] = L(x) + \frac{1}{2B}\,\mathrm{Var}[v_1(x)]. \tag{49}$$

*Proof.* We have

$$\frac{1}{2}\,\mathrm{Var}[v_B(x)] = \frac{1}{2}\,\mathrm{E}[\|v_B(x)\|^2] - \frac{1}{2}[\|\,\mathrm{E}\,v_B(x)\|^2] = \mathrm{E}[L_B(x)] - L(x). \tag{50}$$

Moreover,

$$\mathrm{Var}[v_B(x)] = \frac{1}{B^2}\sum_{i=1}^{B}\mathrm{Var}[v_i(x)] = \frac{1}{B}\,\mathrm{Var}[v_1(x)]. \tag{51}$$

Together, this yields the assertion. $\qquad\square$

Note that the bias is going down linearly with the batch size. In practice, we found that a batch size of $64$ is usually sufficient to make consensus optimization work.

Alternatively, as was also noted by Ferenc Huszár in a blog post on `www.inference.vc`, it is possible to obtain unbiased estimates of $L(x)$ by subtracting $1/(2B-2)$ times the sample variance of the $v_i(x)$ from $L_B(x)$.

We leave a closer investigation of the benefits and disadvantages of this variant of consensus optimization to future research.

**Additional Experimental Results**

Figure 6: Comparison of the results obtained by training different architectures with simultaneous gradient ascent, alternating gradient ascent and consensus optimization: we train a standard DC-GAN architecture (with 4 convolutional layers), a DC-GAN architecture without batch-normalization in neither the discriminator nor the generator, a DC-GAN architecture that additionally has a constant number of filters in each layer and a DC-GAN architecture that additionally uses the Jensen-Shannon objective for the generator that was originally derived in [10], but usually not used in practice as is was found to hamper convergence. While our algorithm struggles with the original DC-GAN architecture (with batch-normalization), it successfully trains all other architectures.

(a) DC-GAN

(b) + no batch-normalization

(c) + constant number of filters in each layer

(d) + Jensen-Shannon objective

Figure 7: Inception score over the number of iterations for the results in Figure 6. To investigate if consensus optimization can alternatively be interpreted as a way of smoothing the discriminator, we also conducted experiments where we removed the regularization term from the generator loss but keep it for the discriminator loss. We call the corresponding algorithm *smoothing optimizer*. While the smoothing optimizer trains a DC-GAN with batch-normalization successfully where consensus optimization fails, it usually performs worse than consensus optimization.

(a) + constant number of filters in each layer

(b) + Jensen-Shannon objective

Figure 8: Inception score over the number of iterations for training a the architecture from Figure 7 with an additional fully connected layer in the discriminator on cifar-10. We observe similar results as in Figure 7, but the smoothing optimizer fails for the architecture with a constant number of filters in each layer where consensus optimization succeeds.

(a) Standard GAN objective     (b) Jensen-Shannon divergence     (c) Indicator divergence

Figure 9: Using consensus optimization to train models with different generator objectives corresponding to different divergence functions: all samples were generated by a DC-GAN model without batch-normalization and with a constant number of filters in each layer. We see that consensus optimization can be used to train GANs with a large variety of divergence functions.

(a) Simultaneous Gradient Ascent     (b) Alternating Gradient Ascent     (c) Consensus Optimization

Figure 10: Samples generated from a model where both the generator and discriminator are based on a DC-GAN [21]. However, we use no batch-normalization, a constant number of filters in each layer and additional RESNET-layers. We see that only our method results in visually compelling results. While simultaneous gradient ascent completely fails to train the model, alternating gradient ascent results in a bad solution. Although alternating gradient ascent does better than simultaneous gradient ascent on this example, there is a significant amount of mode collapse and the results keep changing from iteration to iteration.

(a) Discriminator loss     (b) Generator loss     (c) Inception score

Figure 11: (a) and (b): Comparison of the generator and discriminator loss on a DC-GAN architecture with 4 convolutional layers trained on cifar-10 for consensus optimization (without batch-normalization) and alternating gradient ascent (with batch-normalization). (c): Comparison of the inception score over time which was computed using 6400 samples. We see that on this architecture consensus optimization achieves much better end results.

# Effect of Hyperparameters

(a) 3 convolutional layers

(b) 4 convolutional layers

Figure 12: Effect of the learning rate on training a DC-GAN architecture with alternating gradient ascent. We use RMSProp [25] as an optimizer for all experiments.

(a) 3 convolutional layers

(b) 4 convolutional layers

Figure 13: Effect of the learning rate on consensus optimization. We use the RMSProp-optimizer [25] for all experiments with a regularization parameter $\gamma = 0.1$ for the architecture with 3 convolutional layers and $\gamma = 10$ for the architecture with 4 convolutional layers.

(a) 3 convolutional layers

(b) 4 convolutional layers

Figure 14: Effect of the regularization parameter $\gamma$ on consensus optimization. We use the RMSProp-optimizer [25] with a learning rate of $2 \cdot 10^{-4}$ for all experiments.

## Footnotes

[6]Note that $v'(\bar{x})$ is usually not symmetric and therefore it is possible that $v'(\bar{x})$ is negative semi-definite and invertible but not negative-definite.