[Reviews · NeurIPS 2017]

Reviewer 1



This paper presents a novel analysis of the typical optimization algorithm used in GANs (simultaneous gradient ascent) and identifies problematic failures when the Jacobian has large imaginary components or zero real components. Motivated by these failures, they present a novel consensus optimization algorithm for training GANs. The consensus optimization is validated on a toy MoG dataset as well as CIFAR-10 and CelebA in terms of sample quality and inception score. I found this paper enjoyable to read and the results compelling. My primary concern is the lack of hyperparameter search when comparing optimization algorithms and lack of evidence that the problems identified with simultaneous gradient ascent are truly problems in practice. Major concerns: 1. The constant discriminator/generator loss is very weird. Any ideas on what is going on? Does this hold when you swap in different divergences, e.g. WGAN? 2. Which losses were you using for the generator/discriminator? From Fig 4, it looks like you’re using the modified generator objective that deviates from the minimax game. Do your convergence results hold for non-zero sum games when f and g are different functions, i.e. f != -g? 3. Experiments: You note that simultaneous and alternating gradient descent may require tiny step sizes for convergence. In your experiments did you try tuning learning rates? Additional optimization hyperparameters? Without sweeping over learning rates it seems unfair to say that one optimization technique is better than another. If consensus optimization is more robust across learning rates then that should be demonsrated empirically. 4. Empirical validation of the failures of simultaneous gradient ascent: are zero real components and large imaginary parts really the problem in practice? You never evaluate this. I expected to see empirical validation in the form of plots showing the distribution of eigenvalues of the Jacobian over the course of optimization for simultaneous gradient ascent. Minor concerns: 1. From the perspective of consensus optimization and \gamma being a Lagrange multiplier, I’d expect to see \gamma *adapt* over time. In particular, ADMM-style algorithms for consensus optimization have dynamics on \gamma as well. Have you explored these variants? 2. Many recent approaches to optimization in GANs have proposed penalties on gradients (typically the gradient of the discriminator w.r.t. Its inputs as in WGAN-GP). Are these related to the approach you propose, and if so is another explanation for the success of consensus optimization just that you’re encouraging a smooth discriminator? One way of testing this would be to remove the additional \nabla L(x) term for the generator update, but keep it for the discriminator update. 3. Figure 5 (table comparing architectures and optimizers) is a nice demonstration of the stability of Consensus relative to other approaches, but did you sweep over learning rate/optimization hyperparameters? If so, then I think this would make a strong figure to include in the main text. 4. I had a hard time following the convergence theory section and understanding how it relates to the results we see in practice.

Reviewer 2



This is a very nice, elegant paper on the optimization challenges of GANs and a simple, well motivated fix for those problems. The continuous-time perspective on optimization makes it very clear why minimax problems suffer from oscillations that regular optimization problems do not, and the proposed solution is both theoretically motivated and simple enough to be practical. I anticipate this will be useful for a wider variety of nested optimization problems than just GANs and should get a wide viewing. My only major suggestion for improving the paper would be to provide a wider evaluation, especially with difficult-to-train architectures known for mode collapse. As is, the results in the main paper show comparable performance to vanilla GAN training with much more stable discriminator and generator objectives, but not a major improvement over cases where alternating gradient ascent fails. Also in Figure 5 in the appendix, it's not clear why consensus optimization fails for DCGAN training - a better discussion of that would help. But it generally seems like a promising approach and I look forward to seeing it applied in problems like inverse reinforcement learning and deterministic policy gradients for continuous control.

Reviewer 3



The paper presents an analysis of the optimization of GANS by analyzing the jacobian of the vector field of the zero sum game. A saddle point exists if this jacobian is negative definite . The paper proposes to penalize the norms of gradients with respect to the parameters of the generator and the discriminator in order to ensure such conditions. clarity: The paper is clear and well written and proofs seem correct. Experiments are supportive of the findings. Comments: - penalizing the l_2 norm of the gradient , requires automatic differentiation , which is feasible in tensorflow, but it involves computing the hessian of generator and discriminator losses. Giving the wall clock timing needed to enforce this constraint is important to see the additional overhead needed to enforce stability. - I wonder why not just enforce the l_{infinity} norm of those gradient , by clipping the gradients with respect to the parameter, this is a common practice in recurrent networks. Notice this is different from weight clipping using in WGAN. It would be interesting to compare the two approaches since clipping the gradient is light weight computationally as it does not require automatic differentiation. - Penalizing the gradient requires computing the hessian, hence gradient descent of this penalized loss can be seen as an approximate second order method: the effective gradient is (I - gamma Hessian (theta,phi) ) * (nabla_phi f , nabla_theta g ) under assumption that the eigenvalues of I + gamma H are less then one (this is the assumption also in the paper ), (I+ gamma H)^ (-1) is approximately equal to( I -gamma Hessian ) ( assuming that high order powers of H vanish). which means that penalizing the gradient yields to an approximate newton update (Hessian being here regularized ). It would be interesting to discuss this angle of second order optimization in the paper. My question here is how much do you think second order optimization is important for the convergence of those saddle point problems such as GANS? penalizing the gradient seems to be equivalent to a first order krylov descent https://arxiv.org/pdf/1301.3584.pdf ? since the hessian is computed by automatic differentiation by penalizing the norms of the gradients is it worth seeing how hessian free optimization perform in the GAN context? or adding other powers of the hessian as in krylov subspace descent?